# Pork Liver Pâté Enriched with Persimmon Coproducts: Effect of In Vitro Gastrointestinal Digestion on Its Fatty Acid and Polyphenol Profile Stability

**DOI:** 10.3390/nu13041332

**Published:** 2021-04-17

**Authors:** Raquel Lucas-González, José Ángel Pérez-Álvarez, Manuel Viuda-Martos, Juana Fernández-López

**Affiliations:** IPOA Research Group, Agro-Food Technology Department, Escuela Politécnica Superior de Orihuela, Centro de Investigación e Innovación Agroalimentaria y Agroambiental (CIAGRO), Universidad Miguel Hernández de Elche, 03312 Orihuela, Alicante, Spain; raquel.lucasg@umh.es (R.L.-G.); mviuda@umh.es (M.V.-M.); j.fernandez@umh.es (J.F.-L.)

**Keywords:** lipolysis, coproducts, Rojo Brillante, *Diospyros kaki*, in vitro digestion, plant-food, UV spectra

## Abstract

Agrofood coproducts are used to enrich meat products to reduce harmful compounds and contribute to fiber and polyphenol enrichment. Pork liver pâtés with added persimmon coproducts (3 and 6%; PR-3 and PR-6, respectively) were developed. Therefore, the aim was to study the effect of their in vitro gastrointestinal digestion on: the free and bound polyphenol profile (HPLC) and their colon-available index; the lipid oxidation (TBARs); and the stability of the fatty acid profile (GC). Furthermore, the effect of lipolysis was investigated using two pancreatins with different lipase activity. Forty-two polyphenols were detected in persimmon flour, which were revealed as a good source of bound polyphenols in pâtés, especially gallic acid (164.3 µg/g d.w. in PR-3 and 631.8 µg/g d.w. in PR-6). After gastrointestinal digestion, the colon-available index in enriched pâté ranged from 88.73 to 195.78%. The different lipase activity in the intestinal phase caused significant differences in bound polyphenols’ stability, contributing to increased lipid oxidation. The fatty acids profile in pâté samples was stable, and surprisingly their PUFA content was raised. In conclusion, rich fatty foods, such as pâté, are excellent vehicles to preserve bound polyphenols, which can reach the colon intact and be metabolized by the intestinal microbiome.

## 1. Introduction

Pork liver pâté is a widespread meat product with high nutritional and density value related to protein, hem-iron, and vitamin content [1]. Furthermore, for some sectors of the population, like children, the elderly, or people with swallowing problems, pork liver pâté can provide an excellent alternative to fresh meat, since it provides high-value nutrients in a small portion and does not require effort in chewing. Nevertheless, given the evidence linking daily meat products consumption with colorectal cancer [2,3], their consumption must be reduced to one or two portions per month. This association between meat products and cancer has been attributed, firstly, to an excessive consumption (50 g per day) and, secondly, to the presence of harmful substances [2]. In pork liver pâté, the most probably dangerous substances are N-nitrous compounds and lipolysis oxidative products. Most manufactured pork liver pâtés have the preservative sodium nitrite, which prevents the proliferation of *Clostridium botulinum* and has antioxidant and colorants effects. However, free form (residual nitrite) can react with hemo-pigments and generate N-nitrous compounds such as nitrosamines [4]. Furthermore, all rich fat foods are susceptible to lipid oxidation, above all, if they are rich in iron and are undergoing the mincing and high-temperature process [5,6].

Diverse strategies to reformulate healthy meat products have been studied and developed by both researchers and the meat industry [7,8,9,10]. One of them is the use of agrofood coproducts, such as antioxidants, nitrite reducing agents, and fat replacers, for their fortification with fiber and bioactive compounds [4,11,12,13].

Recently, many investigations have pointed out that polyphenols, especially in their insoluble-bound form, which remains covalent joining the cell wall, could modulate the intestinal microbiome’s composition [14,15]. Furthermore, some authors pointed out the potential of polyphenol-biofortified food as a novel tool for preventing human gut diseases [16]. Many authors have shown that polyphenol mitigates the early stage of colon inflammation [17,18]. In this line, Direito et al. [19] showed that persimmon phenols inhibit colitis and colon cancer cell proliferation in rats. Persimmon consumption has also been linked with reducing the atherogenic process due to, mainly, their ability to bind bile acids and protect plasma lipid from oxidation [20,21,22]. Regarding persimmon coproducts, they are rich in polyphenols, carotenoids, and fiber [23,24,25]. Previous works showed their ability to reduce residual nitrite in pork liver pâté [26]. Moreover, the use of agro-food coproducts stimulates food industry sustainability. However, during digestion, the nutrients’ behavior related to their digestibility, stability, and absorption can change when the food matrix where it is embedded is modified or new compounds are added; consequently, the postprandial response is also expected to be changed [27]. Accordingly, as foods are more than the sum of their parts, it is essential to investigate the interactions and effects of the matrix on the final biological activity of nutrients, because more is not always the better. For this purpose, in vitro digestion methods are useful screening methods to evaluate the effects of food reformulations or fortifications.

In the last decade, many researchers have evaluated the digestive process’s repercussion in developing lipid oxidation and fatty acids bioaccessibility in fresh meat and fish and their derived products [5,28,29,30,31,32,33] and polyphenols’ stability after in vitro digestion [34,35,36,37,38]. However, most research have not considered the recommendations carried out by the harmonized, static in vitro digestion protocol proposed by the Cost-Infogest network about the increase in lipase activity (2000 UL/mL) and the addition of colipase in the intestinal phase when rich fatty foods are digested [39]. Recently, the Cost network has also recommended the use of gastric lipase [40]. The high cost of individual enzymes and colipase vs. pancreatin could explain these facts. Another alternative to achieve high lipase activity, cheaper than the use of individual enzymes and colipase, is to use pancreatin to avoid exocrine pancreatic insufficiency, previously proposed by Calvo-Lerma et al. [41]. Tullberg et al. [42] pointed out that lipolysis activity can be a critical factor in promoting lipid oxidation and releasing fatty acids after in vitro digestion.

Therefore, the aims of the current work were: (a) to determine the stability of free and bound polyphenols in pork liver pâtés enriched with persimmon (cv. “Rojo Brillante”) flour (3 and 6%) obtained from juice coproducts; (b) to evaluate the stability of the free and bound polyphenol profile and their colon available index, the lipid oxidation (TBARs), and the stability of the fatty acid profile during in vitro gastrointestinal digestion of pork liver pâtés enriched with persimmon flour; and (c) to assess the effect of lipolysis in the mentioned determinations (polyphenol and fatty acid and lipid oxidation) using two pancreatins with different lipase activity.

## 2. Materials and Methods

### 2.1. Reagents

Individual polyphenol standard, green tea catechin mix (G-016): ((+)—Catechin, (−)—Catechin-3-gallate, (−)—Epicatechin, (−)—Epicatechin-3-gallate, (−)—Epigallocatechin-3-gallate, (−)—Gallocatechin, (−)—Gallocatechin-3-gallate), pancreatin from porcine pancreas (P1750), pepsin from porcine gastric mucosa (P6887), and bile from bovine and ovine (B8381) were purchased from Merck (Darmstadt, Germany). Methanol, ethyl acetate, acetone, hexane, and acetonitrile with HPLC grade were purchased from PanReac ApliChem (Barcelona, Spain). Kreon 25,000 U was purchased in a pharmacy with the corresponding permission. Six monoglycosides mixture (pelargonidin 3-glucoside, cyanidin 3-glucoside, peonidin 3-glucoside, delphinidin 3-glucoside, petunidin 3-glucoside, and malvidin 3-glucoside), malvidin, and malvidin 3,5-diglucoside were purchased from Biolink Group-Polyphenols AS (Sandnes, Norway). All other reagents were purchased from PanReac ApliChem (Barcelona, Spain).

### 2.2. Sample Preparation

Three pork liver pâté formulations were made following the indications described by Lucas-González et al. [26]: control pâté, without persimmon flour (CP); enriched pâté with 3% persimmon flour obtained from persimmon (cv. Rojo Brillante) juice coproducts (PR-3); and enriched pâté with 6% persimmon flour (PR-6)]. Two different batches of each studied pork liver pâté formulation (1.5 Kg) were made. Chemical composition of the three pork liver pâtés has been previously published by Lucas-González et al. [26].

### 2.3. In Vitro Gastrointestinal Digestion

In vitro digestion method was carried out following the standardized procedure described by Minekus et al. [39] and taking into account the recommendations of Brodkorb et al. [40] about not including amylase in the digestion of rich fatty foods, keeping enzyme solutions in ice, and starting the count of the incubation time when the enzyme was added to samples. In addition, the recommendations to achieve 2000 lipase units per mL (LU/mL) in intestinal fluid instead of using individual pancreatic enzymes and colipase were applied [39]. The addition of pancreatin (Kreon 25,000 U) to treat exocrine pancreatic insufficiency, previously used to study lipolysis by Calvo-Lerma et al. [41], was also applied.

Before digestion, pork liver pâté was left for 10 min at room temperature and then was smashed to obtain a paste. Carefully, 5.000 ± 0.005 g of sample was weighed into a 50 mL falcon tube. Stock digested solutions (oral, gastric, and intestinal) were prepared with the same saline concentration and pH described by Minekus et al. [39] and then were put in a water bath to temper. Then, 5 mL of simulated oral fluid without salivary amylase was added to sample vortex 5 s and put in a bath with holistic agitation (tubes were put in horizontal positions). The sample was incubated for 2 min at 37 °C and constant agitation. After the oral phase, 7.5 mL of gastric stock solution and 0.2 mL of HCl 0.2 M were added to the oral digested sample; when sample pH was 3.00 ± 0.05, 0.5 mL of pepsin (2000 U/mL) and 5 µL of CaCl_2_ were added and made up to 20 mL with water. Then, samples were left for 2 h in the agitation bath.

Two intestinal conditions (C1 and C2) were simulated to study the probable effect on fatty acids, polyphenols stability, and on lipid oxidation, due to the lipase activity of pancreatin: (C1) using a pancreatin with high lipase activity (Kreon 25,000) vs. (C2) pancreatin from porcine pancreas (P1750) with low lipase activity. Figure 1 shows the final activity of lipase in each simulated intestinal phase. In both simulated intestinal conditions, intestinal stock solution, bile acid (10 mM), and ClCa_2_ were prepared and added similarly. The only differences were related to the type of enzyme used and procedure. Therefore, to simulate intestinal condition 1, 11 mL of stock intestinal solution, 0.2 mL of NaOH (2 M), and 2.5 mL of bile solution (dissolved in water) were added to the gastric sample; for adjusting pH, NaOH was added drop by drop until a pH of 7.00 ± 0.05. Then CaCl_2_, pancreatin solution (5 mL/ 100 UP/mL), and water were added. For condition 2, the simulate intestinal fluid was prepared using 3.2 capsules of pancreatin Kreon (to achieve 2000 UL/mL). Therefore, for 10 samples, 32 capsules were mixed with 160 mL to intestinal stock solution for preparing intestinal simulation. In brief, the capsule was opened carefully and all content was added into a glass bottle and mixed with 100 mL of intestinal stock solution. The bottle was closed and put in magnetic agitation to ambient temperature. Approximately, it took 30 min to completely dissolve. When the enzyme was dissolved, it was transferred to a test tube and filled up to 160 mL with the intestinal stock solution (pH 7.00). After the gastric phase, 0.2 mL of NaOH was added to digested gastric samples, followed by bile acid and CaCl_2_. Then, 16 mL of simulate intestinal fluid (2000 UL/mL) was added and pH was adjusted to 7.00 ± 0.05. Water was added to make up 40 mL. Both intestinal conditions were incubated in agitation for 2 h at 37 °C. A blank for each digestive phase (oral, gastric, and both intestinal phases: C1 and C2) was made by replacing the pâté sample for distilled water. For carrying out all determinations, three independent digestions for each pâté formulation were carried out. Figure 1 shows the in vitro gastrointestinal digestion process and the specific treatment of each sample, according to each determination.

### 2.4. Extraction, Identification, and Quantification of Bound and Free Polyphenol Fractions of Both Persimmon Flour and Pork Liver Pâté

#### 2.4.1. Persimmon Flour and Undigested Pork Liver Pâté

Two polyphenolic compound fractions were studied both in Rojo Brillante flour and in undigested and digested pork liver pâté samples. For the undigested samples, the free polyphenol fraction was extracted as described by Pellegrini et al. [43] with some modifications. In brief, 2 g of sample (flour or pâté) were extracted two times, first with 20 mL of aqueous methanol (20:80; v:v) and then with 20 mL of aqueous acetone (30:70; v:v). After the sonication process (30 min), samples were centrifugated (4 °C; 7200× *g*; 10 min), and both supernatants were mixed and evaporated under vacuum. The sample was resuspended in distilled water (5 mL) and passed through a C-18 Sep-Pak cartridge previously activated (water; MeOH; HCl 0,01N). Sugar and other soluble compounds were discarded, and the polyphenols were collected in acidified methanol (MeOH: formic acid; 99:1; v:v) (1.5 mL for pâté and 3 mL for persimmon flour). The pellet was extracted following the procedure described by Mpofu et al. [44] to achieve the bound-insoluble polyphenol fraction. In brief, 50 mL of NaOH 4 M was mixed with the pellet sample and left in the dark for 4 h. Then, the sample was transferred to a beaker, and 25 mL of cold HCl 6 M was added. The pH of samples was monitored, and a HCl 6 M solution was added to the sample until a pH of 2.00 was achieved. After this, samples were centrifugated for 20 min (4 °C; 8000× *g*). Then, 25 mL of ethyl acetate was added to a separatory funnel, and consecutively all supernatant was added. The mixture was shaken for 2 min and was left overnight to ambient temperature. After recovering the upper phase (ethyl acetate + polyphenol compounds), the supernatant was washed twice with ethyl acetate; in total, 70 mL was used. The ethyl acetate was evaporated under vacuum, and the residue was resuspended in methanol (2 mL for pâté and 5 mL for persimmon flour). At the end, samples were passed through a nylon filter (0.45 µL). For detection, twenty microliters of both fractions (free and bound) were injected in a Hewlett-Packard HPLC series 1200 coupled with a C18 column (Mediterranean Sea18, 25× 0.4 cm, particle size 5 μm; Teknokroma, Barcelona, Spain). The HPLC conditions were the same reported by Genskowsky et al. [45]. Gallic acid, 4-hydroxycinnamic acid, ellagic acid, protecatechuic acid, sinapic acid, syringic acid, vanillic acid, vanillin, ferulic acid, caffeic acid, *p*-Coumaric acid, chlorogenic, rosmarinic acid, catechin, epicatechin, gallocatechin gallate, gallocatechin-e-gallate, catechin-3-gallate, epicatechin-3-gallate, epigallocatechin-3-gallate, quercetin, kaempferol, myricetin, rutin, apigenin, luteolin, luteolin-7-glucoside, naringin, hesperidin, neorecitrin, neohesperidin, malvidin, malvidin 3,5-O-di-β-glucopiranoside pelargonidin 3-glucoside, cyanidin 3-glucoside, peonidin 3-glucoside, delphinidin 3-glucoside, petunidin 3-glucoside, and malvidin 3-glucoside, L-tryptophan were injected in HPLC under the same conditions reported for samples. Retention time and UV spectra were used to identify the polyphenols in samples by comparing them with the standard. A calibrate curve of each mentioned standard was used to quantify polyphenols in samples.

#### 2.4.2. Digested Pork Liver Pâté

In the case of the digested pork liver pâté samples, after each phase (oral, gastric, and both intestinal conditions), samples were put in ice and immediately were centrifugated (4 °C; 7200× *g*; 10 min) (Figure 1). For free fraction, supernatant was passed through a previously activated C-18 Sep-Pak cartridge (water; MeOH; HCl 0,01N), and for bound fraction the pellet was used, which was previously lyophilized. Both samples (supernatant and pellet) were processed as previously described in Section 2.4.1.

Colon available index (%) (CAI) was calculated as previously reported by Lucas-González et al. [38] following the Equation (1):(1)$$\%\;Colonavailable\;index=\frac{PF_{i}}{BP_{c}}\times 100\;$$
where:*PF_i_*: polyphenols (individual or total) in bound fraction after intestinal phase (µg/g d.w).*BP_C_*: Bound polyphenol (individual or total) in undigested enriched pâté (µg /g d.w).

### 2.5. Fatty Acid Profile

#### 2.5.1. Fat Extraction

The Folch method was used for fat extraction both in undigested and digested pâté samples [46]. For undigested samples, 5 g were used, while for digested samples (only after both intestinal simulated phase), the full sample was extracted, as recommended by Brodkorb et al. [40]. Therefore, after the in vitro intestinal phases (C1 and C2), 150 mL of chloroform: MeOH was added to 40 mL of the intestinal sample. After 20 min under magnetic agitation, the mixture was put into a separatory funnel, and 20 mL of NaCl (0.9% p/v) was added. The separatory funnel was vigorously shaken for 1 min, and the sample was left under ambient temperature overnight to achieve complete phase separation. Then, the chloroform phase was collected and evaporated under vacuum. If emulsion was formed after the repose time, the sample was centrifuged for 10 min at 6000× *g*.

#### 2.5.2. Methylation of Fatty Acids

Extracted fat (100 µg) was used for the methylation process. In brief, the fat sample was mixed with sodium methoxide (1 mL) and dichloromethane (0.1 mL), and then it was mixed and heated for 10 min in a bath at 90 °C. After cooling in an ice bath, boron trifluoride-methanol solution (14% in methanol) (1 mL) was added and again vortexed and heated under the same conditions previously described. After cooling, water (1 mL) and hexane (0.6 mL) were added to the sample; then, it was heavily shaken. In the end, the upper phase (hexane + fatty acids) was carefully separated, and it was injected in a gas Gas-Chromatographer HP-6890 (Woldbronn, Germany) equipped with a flame ionization detector (FID) and a Suprawax 280 capillary column (30 m × 0.25 μm film thickness × 0.25 mm i.d.; Tecknokroma Barcelona, Spain), following the same conditions reported by Botella-Martínez et al. [47]. The response factors were calculated with fatty acids standards, and they were compared with the retention times of the FAMEs (Supelco 37 component FAME Mix, Bellefonte, PA, USA). All the samples were analyzed in triplicate and the results were expressed as mg fatty acid/100 g fatty acids. The variation of unsaturated fatty acid after gastrointestinal digestion acids was calculated following the Equation (2):(2)$$\%\;Fatty\;acid\;variation=\frac{FA_{id}-FA_{u}}{FA_{u}}\times 10$$
where:*FA_u_*: Fatty acid content in undigested pâté samples (mg/100 g fatty acid).*FA_id_*: Fatty acid content after intestinal digestion (mg/100 g fatty acid).

### 2.6. Lipid Oxidation (TBARs)

Lipid oxidation in undigested and digested pâté (gastric and intestinal phases) was determined following the spectrophotometric TBARs (thiobarbituric acid reactive substances) procedure described by Sobral et al. [32]. For each digested test tube, two aliquots of 400 µL were taken (Figure 1). In total, 6 determinations were analyzed for each sample. A blank for each digestive phase (oral, gastric, and both intestinal phases (C1 and C2)) was made by replacing the digested sample for distilled water. The absorbance was read at 532 nm. Malonaldehyde acid (MDA) quantification was made using a standard curve with 1,1,3,3-tetramethoxypropane, and the results were expressed as µmol MDA per Kg of the sample. For calculating the increase in lipid oxidation, Equation (3) was used:(3)$$\%\;Lipid\;oxidation\;increase=\frac{TBARS_{d}-TBARS_{u}}{TBARS_{u}}\times 100$$
where:*TBARs_u_*: TBARs values in undigested pâté samples (µmol MDA/Kg).*TBARs_d_*: TBARs values in digested (gastric or intestinal phases) pâté samples (µmol MDA/Kg).

### 2.7. Statistical Analysis

Values were expressed as mean ±standard deviation. Two-way ANOVA statistical assays was carried out. The factors used were pâté type (CP, PR-3, and PR-6) and type of sample (undigested, oral digested, gastric digested, intestinal C1 digested, or intestinal C2 digested). Pearson correlation analysis was used to investigate the relations between the fatty acid profile and lipid oxidation values in undigested samples. Tukey’s post host test was used to elucidate significant differences when the *p*-value was <0.05. Statistical analyses were performed using the STAT Graphic Program.

## 3. Results and Discussion

### 3.1. Bound and Free Phenolic Compounds in Persimmon Flour

Among 42 detected polyphenolic compounds in persimmon flour, 37 of them were identified (Table 1), and 12 were confirmed by standards (gallic acid, catechin, caffeic acid, epigallocatechin-3-gallate, gallocatechin-3-gallate, *p*-Coumaric acid, epicatechin-3-gallate, ellagic acid, ferulic acid, myricetin, quercetin, and kaempferol). The others were tentatively identified; we analyzed their absorbance spectrum, consulting bibliography, and considering that glycosylated polyphenols eluded earlier than their aglycone due to the hydroxyl groups’ contribution of the sugar [48]. Regarding tentative identification, except for compounds No. 4, 18, and 33, all the others showed an identical absorbance spectrum to the family or aglycone assigned. Figure 2 shows the absorbance spectrum of some of these compounds. The compound identified as catechin glucoside (Figure 2C) could also be proanthocyanidin dimers, since some of them elude before catechin and present the same major absorbance peaks [49,50]. The absorbance spectrum of the compound identified as coumaric glucoside (No. 4) had differences with the absorbance spectrum of *p*-Coumaric acid (Figure 2A) but had the same maximum absorbance peak. Furthermore, Martínez-Las Heras et al. [50] found coumaric acid-O-hexoside in persimmon leaves, and Sentandreu et al. [51] detected a coumaric acid-pentoside-hexoside in Rojo Brillante flesh. Figure 2A shows the spectrum of compound No. 33, which has been identified as a coumaric acid derivative because it eluded after coumaric acid and presented an absorption peak around 400 nm. The compound No. 18 was identified as anthocyanin, since it presented a very close absorbance spectrum to anthocyanin family as can be observed in Figure 2E. Concerning the unknown polyphenols observed, two of them, No. 12 and 39, showed very close UV spectra to flavone compounds (luteolin and apigenin) and phenolic acids (Figure 2D,F). In the available bibliography, six flavone C-di-hexoxides in Rojo Brillante flesh have been detected by Sentandreu et al. [51]. However, due to the substantial differences in the shape and maximum absorbance spectrum, these compounds have not been identified nor quantified.

The most abundant polyphenol family found in persimmon flour was flavonoids, and their main subfamilies were flavonols and flavanols, with 11 and 10 compounds detected, respectively. Many authors have previously detected quercetin, kaempferol myricetin, and high diversity of their glycosides, being quercetin and kaempferol predominantly in persimmon fruit cv. Rojo Brillante and in other cultivars [25,48,52,53,54]. Except for quercetin glucoside III and kaempferol glucoside II, which were detected in both studied fractions, all flavonol compounds were found in persimmon flour in the free fraction. Conversely, all flavanols identified in the persimmon flour, except catechin, were shown in the bound fraction. These results were expected since persimmon is a fruit rich in condensed tannin compounds resulting from the polymerization of the flavan-3-ols units [22,55], which because of their highly polymerized grade could not be extracted sufficiently from the fruit by ethanol or acetone [56]. Catechin usually is found in persimmon samples in a free form, as has been reported by Jiménez-Sánchez et al. [48], who only found catechin in juice made from the Rojo Brillante persimmon. However, Chen et al. [57] reported the same amount of catechin in free and bound fractions from persimmon leaves. At the same time, Suzuki et al. [58] observed that the amounts of catechin compounds were higher in astringent persimmons (such as Rojo Brillante) than in non-astringent persimmons. Furthermore, other authors have detected other flavanols like catechin glucoside, gallocatechin, catechin gallate, epigallocatechin-3-gallate, procyanidin dimers, or prodelphinidin dimers in the leaves and flesh of persimmon cultivar Rojo Brillante and other astringent cultivars [50,53,59].

The following abundant flavonoid subfamily was flavanone, with five detected compounds: three in the bound fraction and two in free forms. Flavanone compounds have never been detected in persimmon flour derived from juice coproducts; however, Jiménez-Sánchez et al. [48] identified naringin glucoside, hesperidin, and eriocitrin in different Rojo Brillante juices, and Martínez-Las Heras et al. [50] detected naringin in persimmon leaves.

As regard to phenolic acids, all of them were detected in bound form, with the exception of ellagic acid. These results could be expected since phenolic acids usually are found in vegetable foods bound to the cell wall [60]. Three of them were tentatively identified as glucosides. Monogalloyl-hexoside compounds have been mainly reported in persimmon fruit [48,51,54]. Furthermore, Maulidiani et al. [54] detected two vanillin glucosides (1-O-Vanilloyl-beta-D-glucose isomer and 1-O-Vanilloyl-beta-D-glucose) in different persimmon cultivars.

It is relevant to highlight that the main amount of polyphenols, both qualitatively and quantitatively, was found in the bound fraction of persimmon flours at 67% and 95%, respectively (Table 2). Although fruits are richer in free polyphenols than bound ones [60], and Chen et al. [57] in their study about the polyphenols fractions of persimmon leaves showed that the highest polyphenol amounts were in the free fraction followed by bound and conjugate fractions, this result could be expected since the persimmon flour was obtained from coproducts of the juice industry. The gallic acid showed the highest amounts in the bound fraction, representing 72% of the total amount followed by flavanone glucoside IV and catechin glucoside I. Previously the abundance of gallic acid in persimmon flour, fruits, and leaves has been reported [25,50,53,61]. In the free fraction, the predominant compounds were catechin followed by quercetin glucoside III and flavanone glucoside III. The sum of them represented around 50% of the free polyphenols quantified in the persimmon flour.

### 3.2. Bound and Free Phenolic Compounds in Undigested Enriched Pâté

In both undigested enriched pâté samples (PR-3 and PR-6), 21 bound polyphenols and 2 free, provided by persimmon flour since those compounds were not detected in control pâté, were found (Figure 3). Among 21 bound polyphenols compounds detected in enriched pâté samples, 16 could be quantified (Figure 4), and the others correspond to the unknown compounds detected in persimmon flours, compounds No. 6, 12, 36, 39, and 41. Interestingly, a compound identified as ellagic acid glucoside (No 43) (Figure 2G and Figure 3) was detected in enriched pâtés but not in persimmon flour. This fact could be due to an overlap of the compounds since the ellagic acid glucoside elutes at the same retention time and wavelength as that of gallocatechin-3-gallate glucoside, which was detected in persimmon flour (Figure 3). Consequently, if this compound is to be identified in persimmon flour, the HPLC conditions should be changed to separate both compounds and modify their retention time. The ellagic acid was also detected in bound fraction instead of in free faction. This fact could also be due to that previously discussed.

The polyphenol amount quantified in each enriched pâté was in concordance with the theoretical prediction (Table 3) except for gallocatechin glucoside, which showed higher amounts than expected. The sum of both fractions in the PR-3 was 285.93 ± 14.60 µg/g d.w and 791.98 ± 121.64 µg/g d.w in the PR-6; the free fraction only contributed around 1%. Significant differences were observed in the polyphenol amount between both enriched pâtés (*p* < 0.05). As can be expected, in both enriched pâtés, gallic acid was the prominent compound quantified followed by flavanone I and gallocatechin gallate glucoside. However, flavanols were not stable in pâté samples, since from 10 bound flavanols detected in persimmon flour, only the gallocatechin glucoside was seen in pâté samples. These results were different from Ribas-Agustí et al. [49], who observed different flavanols (catechin, epicatechin, epicatechin gallate, epigallocatechin gallate, and three procyanidin: C1, B1, and B2) in dry sausages enriched with extracts from cocoa and grape seeds. Regarding the pâté samples chromatographs (PR-3 and PR-6) for the bound fraction, we saw several peaks at the same retention time of many flavanols detected in the persimmon flours; however, their UV spectra were not available or were distorted. Gallic acid glucoside, quercetin, and kaempferol were not observed in pâté samples; this is probably due to the low concentration present in persimmon flour (Table 2 and Table 3).

Regarding free fraction, only quercetin glucoside III was detected; furthermore, kaempferol glucoside II was also observed in pâté samples. Other authors have shown flavanone compounds in dry-cured sausages with other added fruit extracts [62]. The differences in the current work could be related to the initial polyphenol concentration and the different stability attributed to each compound. For example, the content of polyphenols in meat products with added herbal extract (e.g., *Hyssopus officinalis* and *Borago officinalis*) also was low; some polyphenols were not detected at all due to the lower concentrations (0.5% of meat amount) added [63]. The amount of quercetin glucoside III in the undigested PR-3 and PR-6 was 2.38 ± 1.39 µg/g d.w. and 4.64 ± 2.12 µg/g d.w, respectively. At the same time, the amount for kaempferol glucoside II was 0.44 ± 0.13 µg/g d.w. and 0.51 ± 0.20 µg/g d.w., respectively. The low amount of the other free polyphenols in persimmon flour (Table 2) could explain these results.

Given these results, persimmon flour could be considered a good source of insoluble bound polyphenols, especially gallic acid, to enrich meat products.

### 3.3. Stability of Bound and Free Phenolic Compounds in Enriched Pâté after In Vitro Digestion

Pâté samples (PC, PR-3, and PR-6) after each in vitro digestion phase (oral, gastric, and both intestinal conditions: C1 and C2) were assessed for their content in soluble free and bound forms of polyphenols.

Digestion conditions strongly affected both free and bound flavanol content in enriched pâtés. After oral and gastric digestion, kaempferol glucoside II was not detected in any enriched pâté (PR-3 and PR-6). Quercetin glucoside III was detected, as trace, in PR-3 pâté after both oral and gastric digestion, while in PR-6 pâté, it was quantified at a level of 1.05 ± 0.28 µg/g d.w. after oral digestion and at 1.09 ± 0.03 µg/g d.w after gastric digestion. Both flavonols have been previously observed in digested samples of persimmon flours from Rojo Brillante coproducts [25] in both oral and gastric phases. Furthermore, kaempferol glucoside was lost even more than quercetin glucoside. Other authors have reported lower content of free rutin and isoquercetrin in digested carob flour in the gastric step than in the oral step [36]. In contrast, different quercetin glucosides as quercetin-galloyl-hexoside, quercetin-3-galoctidase, quercetin-3-glucoside, and quercetin-3-xyloside present in lyophilized maqui showed higher amounts in gastric step than oral step [34]. Therefore, the gastric medium can compromise glycosylated flavonols’ stability or increase their content probably due to the food matrix’s release under acid conditions. The intestinal phase data from the free fraction are not available due to problems in obtaining the sample. The high amount of intestinal supernatant, with lots of suspension compounds as fatty acids and proteins, presented problems with passing the supernatant through the column due to column obturation and consequently sample loss. Another procedure needs to be developed to study free fractions on rich fatty and protein foods and low in free polyphenols. Besides, amino acids generate interferences in the chromatograph since many amino acids, especially aromatic ones, have their maximum absorbance at 280 nm, making it challenging to identify polyphenols. The interference in the free fraction of aromatic amino acid after in vitro digestion was recorded by Lucas-González et al. [37] and Podio et al. [38] in cereal-based foods. For this reason, other procedures based on protein precipitation (with TCA or phosphoric acid) were tried. After centrifuging the sample, the supernatant was filtered, and the pellet was hydrolyzed to break down the possible bond between polyphenols and proteins. However, due to the work and resources required to carry out the methodology and the need to improve it, the preliminary results are not shown.

About bound fraction, after the oral phase, the quantified bound compounds in enriched pâté samples (PR-3 and PR-6) showed two different tendencies. On the one hand, some polyphenols (caffeic acid, *p*-Coumaric acid, coumaric acid glucoside, gallic acid (only in PR-3%), ellagic acid, ellagic acid glucoside, vanillin glucoside, kaempferol glucoside II, and quercetin glucoside II) did not show variations with respect to the correspondent undigested sample. On the other hand, the rest of the polyphenols (the three glycosylated flavanones (I, II, and IV), the gallic acid (only in PR-6), the coumaric acid derivative, and the anthocyanin) showed a significant decrease with respect to their initial content (*p* < 0.05). Some of these polyphenols, since the gastric digestion returned them to their initial amount (referred to the undigested sample), therefore showed significant differences with oral values (*p* < 0.05).

The gastric phase caused a dramatic effect on ellagic acid and kaempferol glucoside II, which were not detected after this phase, nor after both intestinal conditions studied (C1 and C2). These results agree with those shown by Chait et al. [36], who also did not see insoluble, bound forms of kaempferol and other flavonols like myricetin after gastric and intestinal digestion in carob flours. Nevertheless, Gullón et al. [35] detected ellagic acid after gastric digestion in pellet fraction of pomegranate peel flour, and their content was higher than in the oral phase. In this line, Colantuono et al. [64] showed higher amount of bound ferulic acid after duodenal phase than after gastric phase in pomegranate peel-enriched cookies. However, considering that pomegranate is rich in ellagitannins [35,64], this increase could be due to the breakdown of ellagic polymers [65]. The content of caffeic acid and ferulic acid was also significantly reduced regarding oral steps in both studied pâtés (PR-3 and PR-6).

Furthermore, these phenolic acids after C1 intestinal phase showed a slight increase concerning gastric phase but not higher than oral or undigested value. The behavior observed in bound ferulic acid was observed in durum wheat spaghetti samples without and with persimmon flours and in carob flour [36,38]. Regarding bound caffeic acid, Juániz et al. [66] reported different behaviors after the intestinal step, depending on the type of treatment carried out on the pepper: dramatic losses in raw and fried (in olive oil) pepper and a small reduction in fried (in sunflower oil) and griddled pepper.

Among 14 polyphenols observed after the intestinal phase in both enriched pâtés (PR-3 and PR-6), 4 of them, flavanone glucoside I and IV, vanillin glucoside, and ellagic acid glucoside, showed higher values after C2 intestinal phase than after C1 intestinal phase (*p* < 0.05). Furthermore, the amounts of gallic acid in both enriched pâtés, of *p*-Coumaric and coumaric acid glucoside in the PR-3 and of caffeic acid and flavanone glucoside II in the PR-6, were similar after both intestinal phases (C1 and C2), showing the highest values after C2 intestinal phase (*p* < 0.05). In the case of gallocatechin glucoside, the behavior was the opposite, showing the highest values in both enriched pâtés after C1 intestinal phase (*p* < 0.05). Therefore, only caffeic acid, ferulic acid, and quercetin glucoside II showed the same quantity after both intestinal conditions (*p* > 0.05).

Given these results, it could seem that C2 intestinal phase, with high lipase activity, is more suitable for recovering polyphenolic compounds after digestion than C2 intestinal phase. This fact could be associated with a high level of free fatty acids on the digestive medium, which could have a protective effect on polyphenols. In this line, results reported by other authors would seem to support this hypothesis. Juániz et al. [66] showed that the presence of oil (olive or sunflower) decreased the loss of bound polyphenols in fried green pepper after in vitro digestion in greater amounts compared with crude and grilled green pepper. Pineda-Vadillo et al. [67] hypothesized that the higher stability of anthocyanin in pancakes and omelets than in milkshakes and custard desserts was related to fatty acid release and the consequent reduction in the pH of the medium. Furthermore, McClements et al. [68] denoted that added fatty acids to the food matrix improved polyphenol bioaccessibility, and Guo et al. [69] showed more bioavailability of quercetin in rich fat diets, in a human study. Thus, although the differences between intestinal conditions 1 and 2 were not observed in all detected polyphenols (Figure 4), they were probably due to the different polarity of polyphenols compounds founded. With these results, high lipase activity (2000 UL/mL) would be recommended when rich fatty foods undergo in vitro digestion to study their polyphenols’ bioaccessibility.

Table 4 shows the colon available index (%) of enriched persimmon flour pâtés. Other authors call this index a bioaccessibility index or recovery index (the way to calculate is the same) [34,36,66,70,71]. However, considering that the polyphenols were observed in the bound fraction, which is not released to medium, they probably will arrive intact to the colon and some of them could be metabolized by the intestinal microbiome. For this reason, it is considered that this name is more appropriate for bound compounds that have been detected in food samples after in vitro digestion.

The following compounds were released from the matrix in both enriched pâtés (PR-3 and PR-6) after intestinal digestion (independently of the intestinal condition used): caffeic acid, ellagic acid glucoside, vanillin glucoside, and gallocatechin glucoside. The lower CAI value was shown for ellagic acid glucoside after C1 intestinal phase, with values around 10%. Other authors have reported similar values for caffeic acid, gallocatechin glucoside, or quercetin glucoside III (Table 4) but different values for other compounds like ferulic acid (27.1%), isoquercitrin (34%), protocatechuic acid (35.28%), or chlorogenic acid (36.8%) [36]. However, in the current work, the CAI of ferulic acid (except for PR-6 after intestinal phase) was higher than 100%, indicating that more compounds were present in the food matrix than previously detected, and the intestinal medium would help to make their extraction easier. This increase was also shown in the gallic acid, *p*-Coumaric acid, coumaric acid glucoside, and flavanone glucoside IV.

These results agree with Adom and Liu’s work [72] reporting that insoluble bound phenols can resist gastrointestinal digestion and reach the colon. Considering that gallic acid and flavanone glucoside IV were the principal polyphenols in enriched pâté samples, this could explain the total CAI % shown, which was upper to 100% except in the PR-6 sample after C1 intestinal digestion, which was around 90%. These results agreed with Huang et al. [71], who reported an increase in bound polyphenols in some seaweed species after in vitro digestion. In contrast, other authors showed a polyphenol bioaccessibility index lower than 100% [36,66,70]. Furthermore, in a previous work where persimmon flour was used to enrich durum wheat spaghetti at the same concentration that was used in the current work (3 and 6%), only two bound polyphenols contributed by persimmon flour (gallic acid and coumaric acid glucoside) were detected, although in lower amounts than in enriched pâté [38]. This could be denoting the effect of the manufacturing process and food matrix on their stability, release, and availability. Other authors have also reported great effects of the food matrix on polyphenol stability [67,73].

The high number of polyphenols in pâté samples resisting enzymatic digestion and remaining covalently joined to the cell wall could arrive to the colon and modulate the intestinal microbiome’s composition [14,15]. Several works have pointed out the relevance of intestinal microbiota in generating polyphenol metabolites, which are better absorbed than the precursor polyphenols, showing more bioactivity and persisting longer in blood [74,75]. About gallic acid, Li et al. [76] demonstrated that in a microbiome-metabolomics analysis in rats with induced colitis and treated with gallic acid, gallic acid intervention attenuated colitis by improving body weight loss, hematochezia, epithelial integrity of colon tissue, oxidative stress, and inflammation in the colon. Furthermore, gallic acid increased beneficial bacteria and decreased pathogenic bacteria. Yang et al. [77], in a review about the impact of gallic acid in gut health, reveled the potential of this acid and its derivatives for the treatment and prevention of gastrointestinal diseases through interaction with the gut microbiome and modulation of the immune response. The modulation of gut microbiota and the immunology response also have been shown in an animal study where female mice with induced chronic inflammation were fed meat product with an added antioxidant extract [18].

### 3.4. Lipid Oxidation in Undigested and Digested Pâté Samples

The lipid oxidation values of three pâté formulations determined as TBARs in undigested and digested pâtés can be observed in Table 5.

These values are in accordance with that reported by Goethals et al. [31] in commercial liver pâté samples. In undigested pâtés, the highest TBARs values were shown in PR-6, followed by PR-3 and PC, showing significant differences (*p* < 0.05). The prooxidant effect of vegetable ingredients, especially paste date coproducts in pork liver pâté, was been previously established by Martín-Sánchez et al. [78]. The food matrix disruption in pork liver pâté enriched with persimmon flour observed by Lucas-González et al. [26], probably due to fiber and sugar, could induce lipid oxidation. Although other authors have supported the antioxidant effect of some rich polyphenol extracts in the meat matrix [32,62,79,80], it seems that the extract composition and meat matrix could influence the antioxidant-prooxidant outcomes [78]. About the effect of digestion on lipid oxidation, results showed that the gastric phase did not have oxidant effect in the samples (Table 5); some values were even lower than in undigested pâté. Still, no significant differences were shown among undigested and gastric samples in pâté samples (*p* > 0.05). Although some authors considered gastric medium as a bioreactor to promote lipid oxidation [81], in these pâté samples, it was not observed. These results could be due to the antioxidant action of ascorbate and nitrate, which were added at 0.5% and 125 ppm, respectively, in pâté formulation but also due to the absence of gastric lipase. Other authors have observed lipid oxidation reduction due to antioxidants compounds. Sobral et al. [32] showed that 0.2% oregano in chicken burgers reduced its TBARs values after intestinal digestion values of 3 nmol/g. Besides, Martini et al. [30] demonstrated that extra-virgin olive oil in low amounts (2.5%) reduces lipid oxidation in grilled turkey breast meat after in vitro co-digestion of both foods. Furthermore, Steppeler et al. [5] reported lower MDA values in minced pork after gastric digestion (6.7 µmol/Kg) than in minced chicken and salmon, pointing out that the presence of polyunsaturated fatty acids was determinant in increasing lipid oxidation.

Although enriched pâtés presented more oxidative end products than the control after the gastric phase (*p* < 0.05), no differences in TBARs values between pâté samples after the study of both intestinal conditions (1 and 2) were shown (*p* > 0.05). This could be due to the antioxidants present in persimmon flours as polyphenols or carotenoids, which would be released from the food matrix under intestinal conditions. It would be supported by the high levels of bound polyphenols detected in enriched pâté samples (Table 4). However, after both intestinal digestions (C1 and C2), lipid oxidation increased significantly (*p* < 0.05) in all samples. These results were concordant with Goethals et al. [31], who subjected three different commercial liver pâtés to in vitro gastrointestinal digestion. This significant rise in lipid oxidation after the intestinal phase could be related to the increase in the lipolysis in the intestinal phase mediated by lipase. It is known that fatty acids are more susceptible to oxidation than triglyceride and can induce pro-oxidation reactions by attracting pro-oxidant metals and co-oxidizing triglycerides [82,83]. In the same way, TBARs values after intestinal phase C2 were higher than after intestinal phase C1 (*p* < 0.05). Therefore, it seems that the increase in lipase activity in the digestive medium significantly increases lipid oxidation. These results were in concordance with Tullberg et al. [42], who studied the effect of lipolysis on lipid oxidation using a lipase inhibitor (Orlistat). They reported that Orlistat significantly reduced lipolysis and MDA formation in marine oils during in vitro gastrointestinal digestion. Furthermore, they reported the increase of primary lipid oxidation products, such as 4-hydroxy-2-hexenal, through the action of gastric lipase. Larsson et al. [28] also reported a relation between lipolysis activity and increased lipid oxidation.

In addition to TBARs values, the behavior of lipid oxidation (comparing initial values: undigested pâté) after gastric and intestinal digestion can be seen in Figure 5. After gastric phase, discrepant values were shown, derived from the fact that similar values were shown among the three undigested pâtés and their respective after-gastric phase. However, after both intestinal phases (conditions 1 and 2), lipid oxidation increased: PR-6 < PR-3 < PC (*p* < 0.05). Regarding results shown by Larsson et al. [28] and Nieva-Echevarría et al. [84], who reported that oxidized oils showed more lipid oxidation than non-oxidized oil after in vitro gastrointestinal digestion, it could be expected that, after intestinal digestion, PR-6 showed the highest TBARs value. Although the initial TBRAs value of the undigested PR-6 was higher than undigested PC, after gastrointestinal digestion, pro-oxidation reactions were not induced by the presence of persimmon flour in pâté samples; on the contrary, lipid oxidation reactions were reduced (Figure 5). Furthermore, the protective effect of persimmon flour on lipid oxidation was concentration-dependent (*p* < 0.05).

### 3.5. Fatty Acid Profile of Pâté and Its Stability after In Vitro Digestion

Fatty acid profile of undigested and digested pâté samples can be observed in Table 6. The main fatty acids in the three studied pâté formulations were oleic acid (C18:1), palmitic acid (C16:0), linoleic acid (C18:2), and stearic acid (C18:0). These four fatty acids represent around 90% of total fatty acids in pâté samples. These results agree with the fatty acid profile of pork meat and pork liver pâté [5,85].

The undigested PC showed the highest values regarding the four observed polyunsaturated fatty acids (PUFAs): linoleic (C18:2), linolenic (C18:3), cis-11,14-eicosadienoic acid (C20:2), and cis-8,11,14-eicosatrienoic acid (C20:3). The monounsaturated fatty acid C20:1 and the saturated fatty acid C12:0 were detected in higher amount in CP than in enriched pâtés (PR-3 and PR-6) (*p* < 0.05). In contrast, both pâtés with persimmon flours (PR-3 and PR-6) showed the most significant quantities of the saturated fatty acids, C10:0, C15:0, and C17:0, and the monounsaturated fatty acids palmitoleic acid (C16:1) and oleic acid (C18:1). Considering that PUFAS are more sensitive to oxidation due to their high double bounds [5,86], it could explain the differences in lipid oxidation shown among pâtés. In this line, a positive and statistic correlation was demonstrated between the amount of the fatty acid C20:3 in undigested pâté samples and their lipid oxidation (R2 = 0.74; *p* < 0.015).

Table 6 shows the fatty acid profile of pâté samples after intestinal digestion. This profile was not qualitatively modified after the digestion process. PUFAs content in the three pâté samples was increased after digestion. The fatty acids C18:3 and C20:3 reported the highest variations with respect to undigested pâté samples (Figure 6). It was an unexpected result since, as mentioned before, gastrointestinal conditions promote lipid oxidation [81,87], and the greater the number of double bonds, the greater the oxidation [86]. Therefore, the expected result would have been a decrease in their content, as was reported by Sobral et al. [32] in digested chicken meat burgers and Liu et al. [33] in digested mushroom *Oudemansiella radicata*. However, both authors reported an increase in PUFA after the cooking process. Liu et al. [33] attributed this effect to the fact that PUFAs are part of the cell membrane and could resist oxidation derived from high temperature. Zhu et al. [29], in a study about the release of fatty acids from emulsified lipids during in vitro digestion, pointed out that fatty acid release is dependent on the structure of triglycerides and on the length of the carbon chain. The tendency observed was the longer the carbon chain, the greater the time needed to release. In another study, Costa et al. [88] showed a lower PUFAs bioaccessibility in grilled salmon than in crude salmon, probably due to the fact that PUFAs are prone to attach to the protein aggregates formed by a result of cross-linking reactions induced by grilling. Therefore, considering these works, it could be hypothesized that part of the PUFA content remains attached to pâté matrix after the extraction process but could be released during the digestion process.

Comparing both intestinal conditions (C1 and C2), significant differences in the amount of some fatty acids were found. In all pâtés (PC, PR-3, and PR-6), the number of fatty acids C18:3 and C14 after C1 intestinal phase was higher than after C2 intestinal phase (*p* < 0.05). In addition, the highest value of C20:3 was found after the C1 intestinal phase, but these differences only were significant in PR-3 (*p* < 0.05). However, fatty acid variations (Figure 6) showed a significant difference among intestinal conditions in all studied pâtés regarding the fatty acid C20:3 (*p* < 0.05). We suppose that these differences between methods are related to different lipolysis activity. The highest lipase activity in C2 intestinal phase promotes the rapid release of fatty acids; consequently, the fatty acids were more time-exposed to the intestinal medium and underwent more lipid oxidation than the fatty acid release under C1 intestinal condition. In contrast, enriched pâtés showed the lowest percentage of C18:1 in C1 intestinal condition (*p* < 0.05). However, although significant differences in the amount of specific fatty acids have been shown between undigested and digested samples, in general, the fatty acid profile in pâté samples after digestion was preserved due to the antioxidant activity of antioxidants used in the formulation, as mentioned before.

Given these results, more studies are needed to understand the mechanism and behavior of fatty acids after digestion as well as food matrix, antioxidants, and lipase activity implications in the oxidation process and their stability in order to generate formulation strategies in rich fatty foods with high nutritional value after the digestive process.

## 4. Conclusions

To the best of our knowledge, this is the first time that the bound and free polyphenolic compounds’ stability has been studied in enriched meat products (with extracts rich in polyphenols) after in vitro gastrointestinal digestion.

Persimmon flour is a good source of bound polyphenols, especially gallic acid and flavanone glucosides, and can be successfully used to enrich pork liver pâté. Rich fatty foods such as pâté are excellent vehicles to preserve bound polyphenols, which could arrive at the colon intact and so be metabolized by the intestinal microbiome. However, PUFAs stability is negatively affected, inducing their oxidation, especially when added at the highest concentrations (6%). The use of two pancreatins with different lipase activity (8 UL/mL vs. 2000 UL/mL) considerably affects both the stability of bound polyphenol compounds and lipid oxidation. The highest number of bound polyphenols and TBARs values was reached after C2 intestinal phase: the higher the rate of lipolysis, the higher the number of fatty acids in the medium, which induced protection of polyphenols against degradation and lipid oxidation. Low variations were shown among fatty acid profiles between undigested and digested pâté samples. Surprisingly, their PUFA content increased after both intestinal phases probably due to the fact that this digestion phase improved their extractability. Lipid oxidation was reduced in pâtés in a dose-dependent way by persimmon flour after both intestinal phases (C1 and C2). Therefore, although the R-6 pâté showed higher oxidation than the control, it was not increased after digestion. Therefore, it could be concluded that lipase activity is an important factor that must be taken into account in the intestinal phase of the in vitro digestion process.

Nevertheless, more studies, both in vitro and in vivo, are needed as well as increasing knowledge about in vitro bile holding ability, lipid digestibility, colonic fermentation, and polyphenols transformations in order to achieve a complete vision of the health implications that could be useful to reinforce the suitability of meat product enrichment with persimmon flour coproducts.

## Figures and Tables

**Figure 1 nutrients-13-01332-f001:**
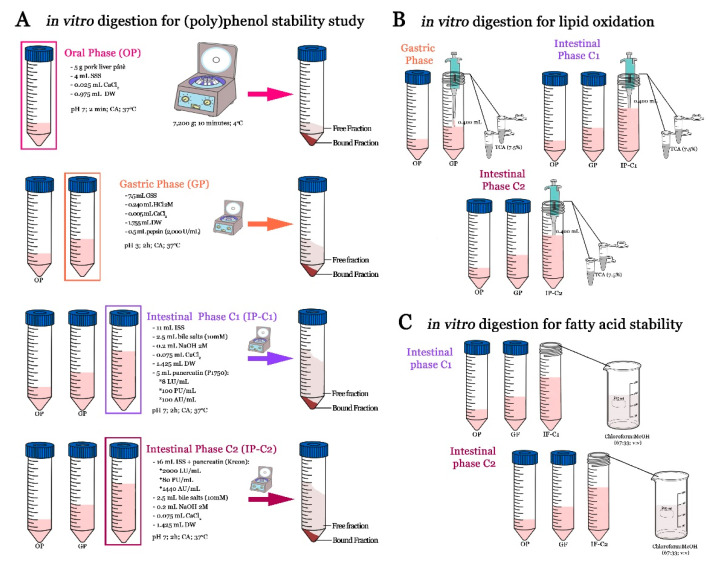
Schematic representation of the in vitro gastrointestinal digestion assays carried out to evaluate (**A**) polyphenol stability; (**B**) lipid oxidation; and (**C**) fatty acid stability in liver pork pâté samples. Imagen caption: SSS: salivary stock solution; GSS: gastric stock solution; ISS: Intestinal stock solution; LU: Lipase units; PU: Protease units; AU: amylase units. TCA: Trichloroacetic acid; DW: Distilled water; CA: Constant agitation.

**Figure 2 nutrients-13-01332-f002:**
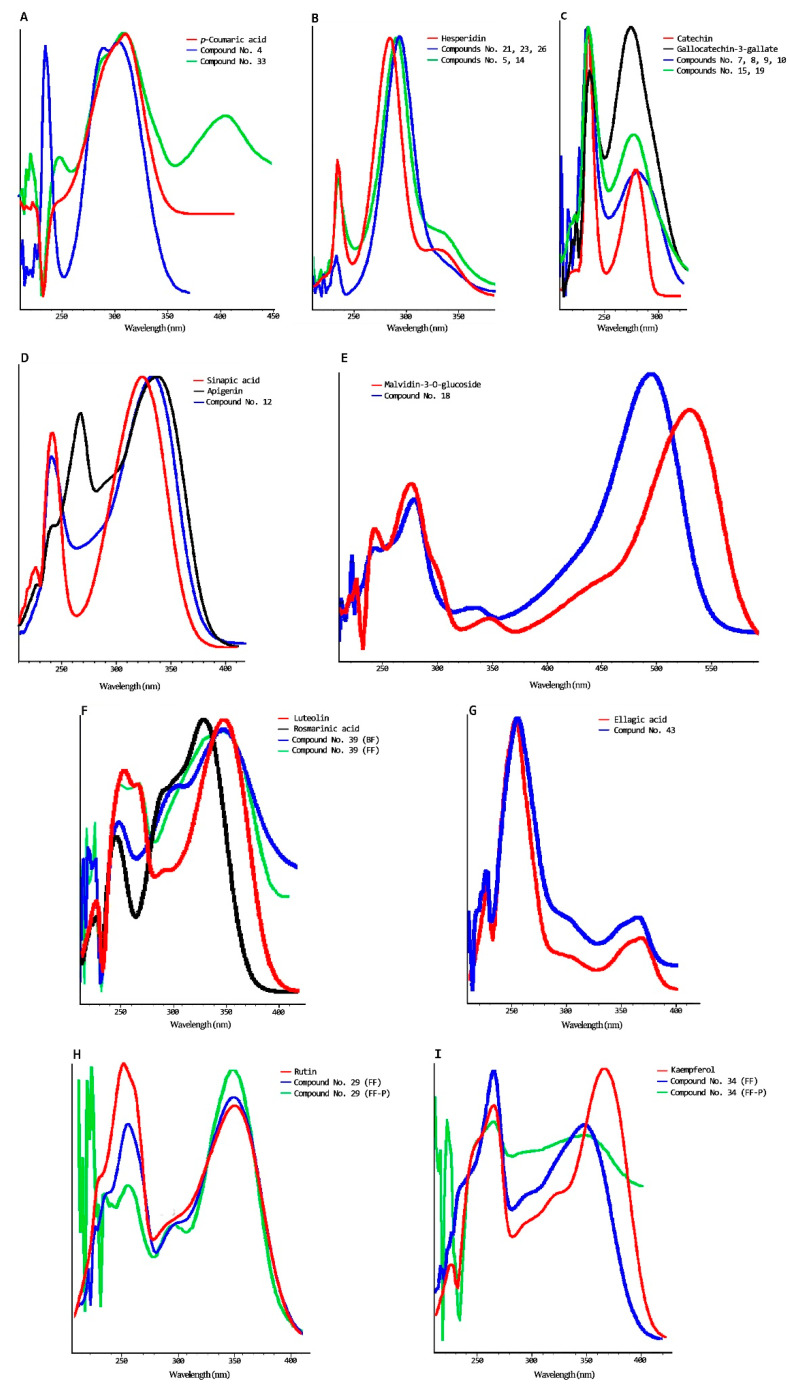
UV spectra of many standard and polyphenols detected in persimmon flour obtained from juice coproducts (cv. Rojo Brillante) and enriched pork liver pâté samples. Image caption: (**A**). *p*-coumaric acid and compounds No. 4 and 33; (**B**). Hesperidin and compounds No. 5, 14, 21, 23, 26; (**C**). Catechin, gallocatechin 3-gallate and compounds No. 7, 8, 9, 10, 15 and 19; (**D**). Sinapic acid, apigenin and compound No. 12; (**E**). Malvidin-3-O-glucose and compound No.18; (**F**). Luteolin, rosmarinic acid and compound No. 39 found in bound and free fractions; (**G**). Ellagic acid and compound No. 43. (**H**). Rutin and compound No. 29 found in persimmon flour and enriched pâté in the free fraction; (**I**). Kampferol and compound No 34 found in persimmon flour and enriched pâté in the free fraction.BF: Bound fraction; FF: free fraction; FF-P: polyphenol detected in the free fraction of the enriched pâtés.

**Figure 3 nutrients-13-01332-f003:**
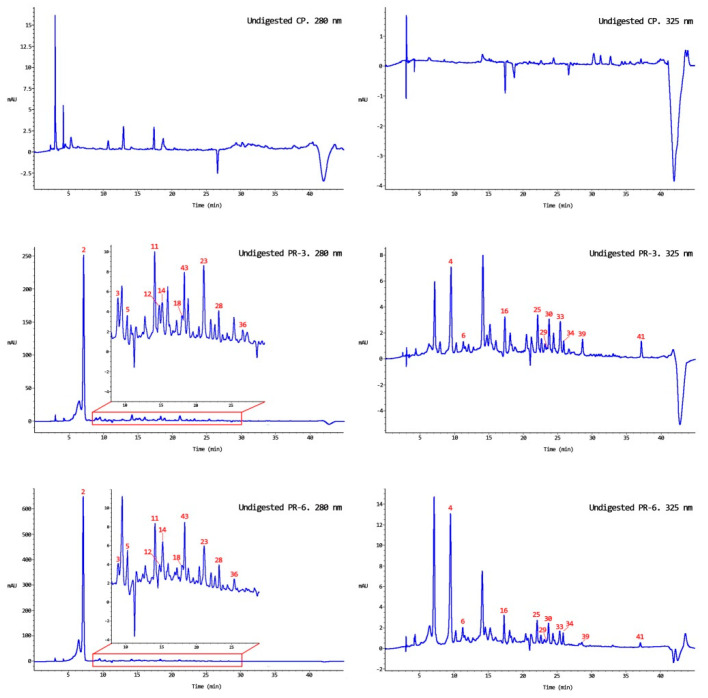
Chromatograms of bound polyphenols in the undigested pork liver pâté samples. CP: control pâté; PR-3 pâté with 3% persimmon flour (Rojo Brillante); PR-6 pâté with 6% persimmon flour (Rojo Brillante); mAU: milli-absorbance unit.

**Figure 4 nutrients-13-01332-f004:**
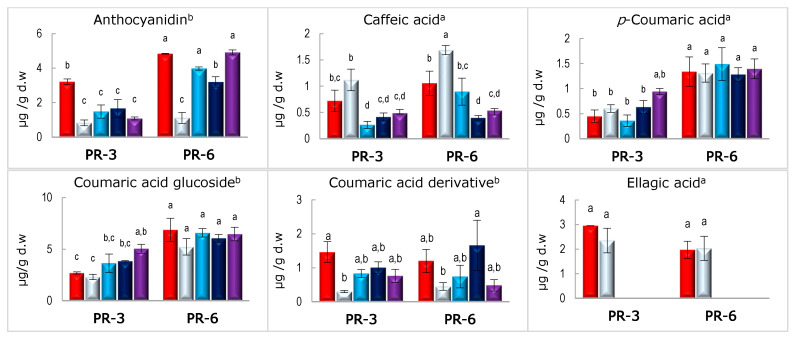
Bound polyphenolic compounds in undigested and digested enriched pork liver pâté. PR-3 pâté with 3% persimmon flour (Rojo Brillante); PR-6 pâté with 6% persimmon flour (Rojo Brillante). **^a^** compound confirmed by standard; **^b^** compound with same UV spectrum of tentative compound or family compound; values with different letter above the bars (**a**–**d**) indicates significant differences (*p* < 0.05) according to Tukey’s post host test.

**Figure 5 nutrients-13-01332-f005:**
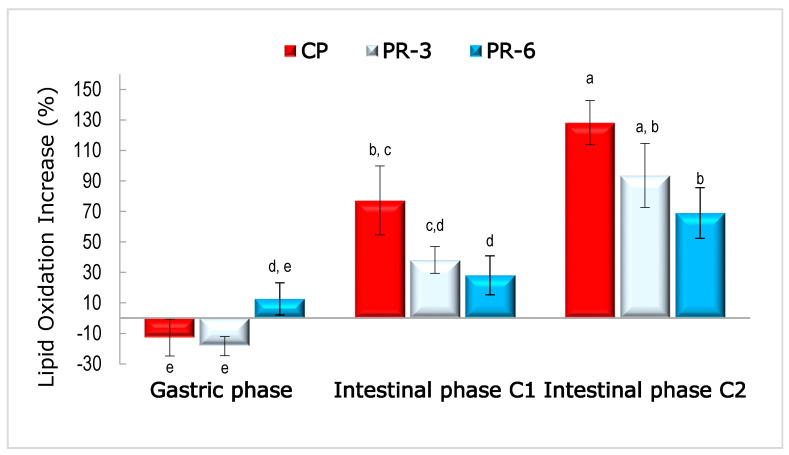
Lipid oxidation increase in pork liver pâté samples after gastric and both simulated intestinal digestions. CP: control pâté; PR-3 pâté with 3% persimmon flour (Rojo Brillante); PR-6 pâté with 6% persimmon flour (Rojo Brillante). Values with different letter indicates significant differences (*p* < 0.05), according to Tukey’s post host test.

**Figure 6 nutrients-13-01332-f006:**
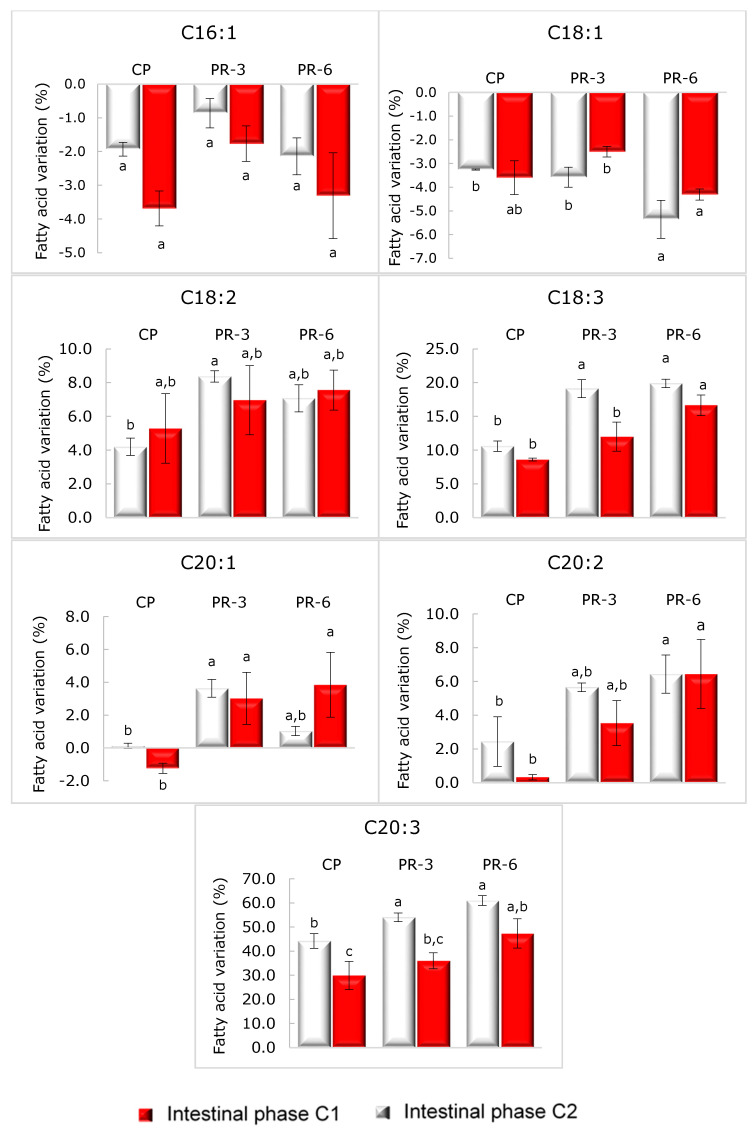
Unsaturated fatty acid variations of pork liver pâté samples. CP: Control pork liver pâté; PR-3: Pork liver pâté with 3% Rojo Brillante flour; PR-6: Pork liver pâté with 6% Rojo Brillante flour after digestion. Values with different letter indicates significant differences (*p* < 0.05), according to Tukey’s post host test.

**Table 1 nutrients-13-01332-t001:** Specification to polyphenol compounds detected in persimmon flour obtained from juice coproducts (Rojo Brillante cultivar).

No	Rt (min)	ʎmax (nm)	Tentative Identification	Fr.	Standard Use to Quantify
1	4.2	234/268	Gallic acid glucoside ^b^	B	Gallic acid
2	6.9	234/270	Gallic acid ^a^	B	Gallic acid
3	8.7	236/270	Gallocatechin glucoside ^b^	B	Gallocatechin gallate
4	9.5	236/300	Coumaric acid glucoside ^c^	B	*p*-Coumaric acid
5	10.2	236/288sh338	Flavanone glucoside I ^b^	B	Hesperidin
6	11.2	236/sh274/344/sh456	Unknow	B	Not quantified
7	12.3	236/280	Catechin glucoside I ^b^	B	Catechin
8	12.8	236/280	Catechin glucoside II ^b^	B	Catechin
9	13.1	236/280	Catechin glucoside III ^b^	B	Catechin
10	13.5	236/280	Catechin glucoside VI ^b^	B	Catechin
11	14.0	236/280/330	Vanillin glucoside ^b^	B	Vanillin
12	14.5	244sh286/334	Unknow	B	Not quantified
13	15.0	236/280	Catechin ^a^	F	Catechin
14	15.2	236/284sh338	Flavanone glucoside II ^b^	B	Hesperidin
15	16.9	236/278	Epigallocatechin-3-gallate glucoside ^b^	B	Epigallocatechin-3-gallate
16	17.3	246 sh298 324	Caffeci acid ^a^	B	Caffeic acid
17	17.8	236/278	Epigallocatechin-3-gallate ^a^	B	Epigallocatechin-3-gallate
18	17.9	242/272/sh334/494	Anthocyanin ^c^	B	Malvidin-3-O-glucose
19	18.5	236/278	Gallocatechin-3-gallate glucoside ^b^	B	Gallocatechin-3-gallate
20	19.2	236/278	Gallocatechin-3-gallate ^a^	B	Gallocatechin-3-gallate
21	19.9	236/288sh334	Flavanone glucoside III ^b^	F	Hesperidin
22	20.7	252/358	Quercetin glucoside I ^b^	F	Rutin
23	21.0	236/288sh334	Flavanone IV ^b^	B	Hesperidin
24	21.2	264/360	Quercetin glucoside II ^b^	F	Rutin
25	22.0	240/sh300/310	*p*-Coumaric acid ^a^	B	*p*-Coumaric acid
26	22.6	238/286sh334	Flavanone glucoside V ^b^	F	Hesperidin
27	22.6	236/278	Epicatechin-3-gallate ^a^	B	Epicatechin-3-gallate
28	23.2	254/362	Ellagic acid ^a^	F	Ellagic acid
29	23.5	258/358	Quercetin glucoside III ^b^	F/B	Rutin
30	23.7	242/sh296/324	Ferulic acid ^a^	B	Ferulic acid
31	24.0	262/358	Quercetin glucoside IV ^b^	F	Rutin
32	25.1	266/348	Kaempferol glucoside I ^b^	F	Kaempferol
33	25.5	240/sh290/308/400	*p*-Coumaric acid derivative ^c^	B	*p*-Coumaric acid
34	25.9	266/350	Kaempferol glucoside II ^b^	F/B	Kaempferol
35	26.4	268/350	Kaempferol glucoside III ^b^	F	Kaempferol
36	26.6	246/260/300	Unknow	B	Not quantified
37	27.6	266/344	Kaempferol glucoside IV ^b^	F	Kaempferol
38	28.5	254/372	Myrecetin ^a^	F	Myrecetin
39	28.8	238sh302/346	Unknow	F/B	Not quantified
40	33.2	256/370	Quercetin ^a^	F/B	Quercetin
41	37.1	254sh264/312sh374	Unknow	B	Not quantified
42	37.8	258/372	Kaempferol ^a^	F	Kaempferol

rt: retention time; Fr.: Fraction; B: bound; F: free; ^a^ compound confirmed by standard; ^b^ Compund with same UV spectrum of tentative compound or family compound; ^c^ Compound with similar UV spectrum of tentative compound or family compound.

**Table 2 nutrients-13-01332-t002:** Free and Bound polyphenolic compounds (µg/g dry weight) of the persimmon flour.

Family	No.	Compound	Free Fraction	Bound Fraction
Phenolic acids	16	Caffeic acid		23.80 ± 4.09
	25	*p*-Coumaric acid		39.40 ± 6.06
	4	Coumaric acid glucoside		77.35 ± 5.37
	33	Coumaric acid derivative		15.71 ± 3.93
	28	Ellagic acid	7.30 ± 1.88	
	30	Ferulic acid		21.08 ± 3.65
	2	Gallic acid		10,074.64 ± 1049.51
	1	Gallic acid glucoside		6.24 ± 0.56
	11	Vanillin glucoside		80.93 ± 25.09
Total phenolic acids			7.30 ± 1.88	10,339.15 ± 1098.26
Flavanones	5	Flavanone glucoside I		221.08± 50.98
	14	Flavanone glucoside II		222.41 ± 15.49
	21	Flavanone glucoside III	28.21 ± 6.74	
	23	Flavanone glucoside IV		1233.34 ± 104.81
	26	Flavanone glucoside V	23.65 ± 3.38	
Total flavanones			51.86 ± 10.12	1676.83 ± 171.29
Flavanols	13	Catechin	37.10 ± 5.42	
	7	Catechin glucoside I		540.52 ± 46.86
	8	Catechin glucoside II		122.69 ± 4.75
	9	Catechin glucoside III		158.00 ± 39.70
	10	Catechin glucoside IV		133.59 ± 32.04
	27	Epicatechin-3-gallate		30.84 ± 2.16
	17	Epigallocatechin-3-gallate		38.81 ± 6.37
	3	Gallocatechin glucoside		327.24 ± 1.23
	20	Gallocatechin-3-gallate		34.17 ± 8.85
	19	Gallocatechin-3-gallate glucoside		50.00 ± 9.49
	15	Epigallocatechin-3-galato glucoside		173.24 ± 43.80
Total flavanols			37.10 ± 5.42	1628.47 ± 185.46
Flavonols	42	Kaempferol	2.96 ± 1.29	
	32	Kaempferol glucoside I	4.03 ± 0.34	
	34	Kaempferol glucoside II	11.84 ± 1.10	9.94 ± 2.19
	35	Kaempferol glucoside III	2.47 ± 0.48	
	37	Kaempferol glucoside IV	2.00 ± 0.64	
	40	Quercetin	7.40 ± 5.64	7.67 ± 1.24
	22	Quercetin glucoside I	6.07 ± 0.66	
	24	Quercetin glucoside II	6.30 ± 0.47	
	29	Quercetin glucoside III	29.76 ± 4.82	32.73 ± 1.02
	31	Quercetin glucoside IV	10.74 ± 0.11	
	38	Myricetin	3.61 ± 0.54	
Total flavonols			87.18 ± 13.09	50.34 ± 4.46
Anthocyanins	18	Anthocyanin		76.67 ± 6.59
Total flavonoids			176.14 ± 28.63	3432.31 ± 367.79
Total			183.44 ± 30.51	13,958.7 ± 1489.2

**Table 3 nutrients-13-01332-t003:** Theoretical number of polyphenols in enriched pork liver pâté samples calculated based on their mean amount in persimmon flour. Values expressed as µg/g d.w.

Bound Polyphenols	PR-3	PR-6	Free Polyphenols	PR-3	PR-6
Gallic	302.2	604.5	Catechin	1.11	2.23
Flavanone glucoside IV	37.0	74.0	Quercetin glucoside III	0.89	1.79
Catechin glucoside I	16.2	32.4	Flavanone glucoside III	0.85	1.69
Gallocatechin glucoside	9.8	19.6	Flavanone glucoside V	0.71	1.42
Flavanone glucoside II	6.7	13.3	Kaempferol glucoside II	0.36	0.71
Flavanone glucoside I	6.6	13.3	Quercetin glucoside IV	0.32	0.64
Epigallocatechin-gallate glucoside	5.2	10.4	Ellagic acid	0.22	0.44
Catechin glucoside III	4.7	9.5	Hydroxycinnamic acid	0.09	0.18
Catechin glucoside IV	4.0	8.0	Quercetin	0.22	0.44
Catechin glucoside II	3.7	7.4	Quercetin glucoside II	0.19	0.38
Vanillin glucoside	2.4	4.9	Quercetin glucoside I	0.18	0.36
Coumaric acid glucoside	2.3	4.6	Kaempferol glucoside I	0.12	0.24
Anthocyanin	2.3	4.6	Myricetin	0.11	0.22
Gallocatechin-3-gallate glucoside	1.5	3.0	Kaempferol	0.09	0.18
p coumaric acid	1.2	2.4	Kaempferol glucoside III	0.07	0.15
Epigallocatechin-3-gallate	1.2	2.3	Kaempferol glucoside IV	0.06	0.12
Gallocatechin-3-gallate	1.0	2.1			
Quercetin glucoside III	1.0	2.0			
Epicatechin-3-gallate	0.9	1.9			
Caffeic acid	0.7	1.4			
Ferulic acid	0.6	1.3			
Coumaric acid derivative	0.5	0.9			
Kaempferol glucoside II	0.3	0.6			
Quercetin	0.2	0.5			
Gallic acid glucoside	0.2	0.4			

PR-3 pâté with 3% persimmon flour (Rojo Brillante); PR-6 pâté with 6% persimmon flour (Rojo Brillante).

**Table 4 nutrients-13-01332-t004:** Colon Available index (%) of bound polyphenolic compounds in enriched pork liver pâté samples.

	PR-3		PR-6	
	Colon Available Index (%)		Colon Available Index (%)	
Polyphenolic Compound	IP-C1	IP-C2	IP-C1	IP-C2
Caffeic acid	57.53 ± 11.25 ^a^	67.30 ± 2.10 ^a^	37.58 ± 4.09 ^b^	44.47 ± 1.52 ^b^
*p*-Coumaric acid	140.7 ± 29.0 ^b^	210.6 ± 13.2 ^a^	95.74 ± 9.99 ^b^	104.06 ± 14.73 ^b^
Coumaric acid glucoside	198.75 ± 2.31 ^b^	268.09 ± 24.94 ^a^	95.26 ± 6.35 ^a^	102.20 ± 11.03 ^a^
Coumaric acid derivative	68.90 ± 11.43 ^b^	52.08 ± 13.43 ^b^	138.36 ± 61.75 ^a^	40.12 ± 13.88 ^b^
Ellagic acid	0.00 ± 0.00	0.00 ± 0.00	0.00 ± 0.00	0.00 ± 0.00
Ellagic acid glucoside	9.98 ± 4.15 ^b^	55.44 ± 2.05 ^a^	15.73 ± 4.08 ^b^	57.73 ± 13.34 ^a^
Ferulic acid	143.54 ± 22.05 ^a^	184.50 ± 5.16 ^a^	58.32 ± 13.66 ^b^	91.87 ± 25.36 ^b^
Gallic acid	176.25 ± 7.29 ^b^	261.66 ± 20.04 ^a^	91.56 ± 2.67 ^c^	109.52 ± 4.58 ^c^
Vanillin glucoside	32.54 ± 2.63 ^b^	87.09 ± 4.90 ^a^	42.34 ± 1.32 ^b^	76.67 ± 12.59 ^a^
Flavanone glucoside I	63.88 ± 18.17 ^b^	95.84 ± 3.31 ^a^	61.15 ± 9.97 ^b^	108.08 ± 9.65 ^a^
Flavanone glucoside II	80.20 ± 29.50 ^a^	113.51 ± 9.32 ^a^	55.13 ± 5.73 ^a^	71.31 ± 18.62 ^a^
Flavanone glucoside IV	115.5 ± 7.0 ^c^	214.1 ± 3.9 ^a^	108.68 ± 11.70 ^c^	173.01 ± 4.46 ^b^
Gallocatechin gallate glucoside	39.53 ± 8.22 ^a^	26.38 ± 2.67 ^a^	45.27 ± 6.83 ^a^	33.59 ± 4.49 ^a^
Quercetin glucoside III	50.283 ± 18.448 ^a^	36.80 ± 0.57 ^a^	41.51 ± 4.51 ^a^	17.43 ± 1.49 ^a^
Kaempferol glucoside II	0.00 ± 0.00	0.00 ± 0.00	0.00 ± 0.00	0.00 ± 0.00
Anthocyanidin	51.8 ± 16.0 ^bc^	33.7 ± 2.8 ^c^	66.17 ± 6.26 ^b^	101.61 ± 3.09 ^a^
Total phenolic acids	170.25 ± 6.91 ^b^	253.98 ± 19.03 ^a^	91.15 ± 2.35 ^c^	108.99 ± 4.35 ^c^
Total flavonols	50.28 ± 18.448 ^a^	36.80 ± 0.57 ^a^	41.51 ± 4.51 ^a^	17.43 ± 1.49 ^a^
Total flavanone	105.80 ± 10.04 ^c^	194.96 ± 1.28 ^a^	91.70 ± 7.57 ^c^	139.57 ± 4.82 ^b^
Total flavan-3-ols	39.53 ± 8.22 ^a^	26.38 ± 2.67 ^a^	45.27 ± 6.83 ^a^	33.59 ± 4.49 ^a^
Total anthocyanidins	51.8 ± 16.0 ^bc^	33.7 ± 2.8 ^c^	66.17 ± 6.26 ^b^	101.61 ± 3.09 ^a^
Total polyphenols	131.39 ± 5.49 ^b^	195.78 ± 12.46 ^a^	88.73 ± 1.55 ^d^	108.74 ± 4.08 ^c^

IP-C1 intestinal phase C1; IP-C2 intestinal phase C2PR-3 pâté with 3% persimmon flour (Rojo Brillante); PR-6 pâté with 6% persimmon flour (Rojo Brillante). Values with different letter in the same row indicate significant differences (*p* < 0.05), according to Tukey’s post host test.

**Table 5 nutrients-13-01332-t005:** Lipid oxidation (µmoles MDA/Kg pâté) of the undigested and digested pork liver pâté samples.

	Undigested	Gastric Phase	Intestinal Phase C1	Intestinal Phase C2
CP	3.19 ± 0.33 ^d^	2.54 ± 0.48 ^d^	5.33 ± 0.68 ^b^	7.80 ± 0.50 ^a^
PR-3	4.20 ± 0.33 ^c,d^	3.52 ± 0.40 ^c,d^	5.53 ± 0.35 ^b^	8.08 ± 0.88 ^a^
PR-6	4.63 ± 0.42 ^b,c^	4.86 ± 0.45 ^b,c^	5.77 ± 0.58 ^b^	8.66 ± 0.85 ^a^

CP: control pâté; PR-3 pâté with 3% persimmon flour (Rojo Brillante); PR-6 pâté with 6% persimmon flour (Rojo Brillante). Values with different letter in the same row indicates significant differences (*p* < 0.05), according to Tukey’s post host test.

**Table 6 nutrients-13-01332-t006:** Fatty acid profile of the undigested and digested pork liver pâté samples.

Fatty Acids (FA), g/100 g of FA	PC			PR-3			PR-6		
	Undigested	IP-C1	IP-C2	Undigested	IP-C1	IP-C2	Undigested	IP-C1	IP-C2
C10:0	0.07 ± 0.00 ^b^	0.07 ± 0.00 ^b^	0.07 ± 0.00 ^b^	0.08 ± 0.00 ^a^	0.07 ± 0.00 ^a,b^	0.07 ± 0.01 ^a^	0.08 ± 0.00 ^a^	0.08 ± 0.00 ^a^	0.08 ± 0.00 ^a^
C12:0	0.10 ± 0.00 ^a^	0.10 ± 0.00 ^a^	0.10 ± 0.00 ^a^	0.09 ± 0.00 ^b^	0.09 ± 0.00 ^b^	0.09 ± 0.00 ^b^	0.09 ± 0.00 ^b^	0.09 ± 0.00 ^b^	0.09 ± 0.00 ^b^
C14:0	1.31 ± 0.00 ^a^	1.30 ± 0.00 ^a^	1.28 ± 0.00 ^b^	1.33 ± 0.01 ^a^	1.32 ± 0.00 ^a^	1.30 ± 0.02 ^b^	1.30 ± 0.04 ^a^	1.30 ± 0.00 ^a^	1.28 ± 0.02 ^b^
C15:0	0.07 ± 0.00 ^c^	0.07 ± 0.00 ^c^	0.07 ± 0.00 ^c^	0.08 ± 0.00 ^b^	0.08 ± 0.00 ^b^	0.08 ± 0.00 ^b^	0.09 ± 0.00 ^a^	0.09 ± 0.00 ^a^	0.08 ± 0.00 ^a,b^
C16:0	22.59 ± 0.50 ^a^	22.68 ± 0.04 ^a^	22.40 ± 0.15 ^a^	23.26 ± 0.39 ^a^	22.59 ± 0.03 ^a^	22.75 ± 0.07 ^a^	22.62 ± 0.72 ^a^	22.52 ± 0.10 ^a^	22.53 ± 0.06 ^a^
C16:1	2.22 ± 0.01 ^b^	2.18 ± 0.00 ^b^	2.16 ± 0.03 ^b^	2.47 ± 0.03 ^a^	2.45 ± 0.01 ^a^	2.43 ± 0.01 ^a^	2.42 ± 0.07 ^a^	2.42 ± 0.01 ^a^	2.39 ± 0.03 ^a^
C17:0	0.39 ± 0.01 ^c^	0.39 ± 0.00 ^c^	0.38 ± 0.00 ^c^	0.43 ± 0.01 ^b^	0.42 ± 0.00 ^b^	0.42 ± 0.00 ^b^	0.46 ± 0.01 ^a^	0.46 ± 0.00 ^a^	0.46 ± 0.00 ^a^
C18:0	11.98 ± 0.44 ^a^	12.24 ± 0.03 ^a^	11.94 ± 0.13 ^a^	12.12 ± 0.47 ^a^	11.61 ± 0.02 ^a^	11.71 ± 0.12 ^a^	11.57 ± 0.35 ^a^	11.58 ± 0.10 ^a^	11.78 ± 0.20 ^a^
C18:1	41.16 ± 0.43 ^b, c^	39.82 ± 0.01 ^b,c^	39.68 ± 0.29 ^c^	42.45 ± 1.29 ^a,b^	40.93 ± 0.18 ^b,c^	42.02 ± 0.90 ^a,b^	42.16 ± 1.92 ^a^	41.06 ± 0.35 ^b^	41.51 ± 0.10 ^a,b^
C18:2 (n 6,9)	13.43 ± 0.36 ^b^	13.99 ± 0.07 ^a^	14.14 ± 0.28 ^a, b^	11.92 ± 0.15 ^d^	12.92 ± 0.04 ^c^	12.75 ± 0.24 ^c^	11.49 ± 0.31 ^d^	12.54 ± 0.09 ^c^	12.59 ± 0.14 ^c^
C18:3 (n 3,6,9)	0.71 ± 0.03 ^b^	0.79 ± 0.01 ^a^	0.75 ± 0.02 ^b^	0.64 ± 0.01 ^c^	0.76 ± 0.01 ^a, b^	0.72 ± 0.01 ^b^	0.63 ± 0.02 ^c^	0.77 ± 0.00 ^a, b^	0.75 ± 0.01 ^b^
C20:0	0.16 ± 0.00 ^b^	0.16 ± 0.00 ^b^	0.17 ± 0.00 ^b^	0.18 ± 0.01 ^a^	0.17 ± 0.00 ^a,b^	0.18 ± 0.01 ^a^	0.19 ± 0.01 ^a^	0.18 ± 0.00 ^a^	0.20 ± 0.01 ^a^
C20:1	0.92 ± 0.01 ^a^	0.92 ± 0.00 ^a^	0.91 ± 0.00 ^a^	0.86 ± 0.01 ^b^	0.89 ± 0.00 ^a^	0.89 ± 0.01 ^a,b^	0.85 ± 0.03 ^b^	0.88 ± 0.00 ^a,b^	0.90 ± 0.02 ^a^
C20:2 (n 11,14)	0.62 ± 0.01 ^a^	0.63 ± 0.01 ^a^	0.61 ± 0.00 ^a^	0.55 ± 0.00 ^c^	0.58 ± 0.00 ^b^	0.57 ± 0.01 ^b,c^	0.52 ± 0.01 ^c^	0.56 ± 0.01 ^b^	0.56 ± 0.01 ^b^
C20:3 (n 8,14,17)	0.57 ± 0.05 ^c^	0.82 ± 0.01 ^a^	0.74 ± 0.03 ^a^	0.52 ± 0.02 ^c^	0.81 ± 0.01 ^a^	0.71 ± 0.02 ^b^	0.51 ± 0.03 ^c^	0.83 ± 0.01 ^a^	0.76 ± 0.03 ^a^
Others	3.71 ± 0.24 ^a^	3.86 ± 0.16 ^a^	4.62 ± 0.74 ^a^	3.01 ± 0.94 ^a^	4.30 ± 0.27 ^a^	3.30 ± 0.79 ^a^	2.69 ± 0.99 ^a^	4.66 ± 0.69 ^a^	4.03 ± 8.03 ^a^
SFA	36.66 ± 0.97 ^a^	37.00 ± 0.07 ^a^	36.38 ± 0.30 ^a^	37.57 ± 0.88 ^a^	36.36 ± 0.06 ^a^	36.61 ± 0.22 ^a^	36.39 ± 1.14 ^a^	36.29 ± 0.21 ^a^	36.50 ± 0.29 ^a^
UFA	59.62 ± 0.91 ^a^	59.15 ± 0.12 ^a^	58.99 ± 0.67 ^a^	59.42 ± 1.51 ^a^	59.34 ± 0.25 ^a^	60.09 ± 1.21 ^a^	58.60 ± 2.38 ^a^	59.05 ± 0.48 ^a^	59.47 ± 0.34 ^a^
MFA	44.30 ± 0.45 ^b^	42.92 ± 0.02 ^c^	42.75 ± 0.33 ^c^	45.78 ± 1.33 ^a^	44.27 ± 0.19 ^b^	45.34 ± 0.93 ^b^	45.44 ± 2.02 ^a^	44.36 ± 0.36 ^b^	44.81 ± 0.15 ^a^
PFA	15.32 ± 0.46 ^b^	16.23 ± 0.10 ^a^	16.25 ± 0.34 ^a^	13.64 ± 0.18 ^c^	15.07 ± 0.06 ^b^	14.75 ± 0.28 ^b^	13.16 ± 0.37 ^c^	14.69 ± 0.11 ^b^	14.66 ± 0.19 ^b^

CP: control pâtè; PR-3 pâtè with 3% of persimmon flour (Rojo Brillante); PR-6 pâtè with 6% of persimmon flour (Rojo Brillante). IP: Intestinal phase. SFA: saturated fatty acid; UFA: Unsaturated fatty acid; MFA: Monounsaturated fatty acid; PFA: Polyunsaturated fatty acid; Values with different letter in the same row (a–c) indicates significant differences (*p* < 0.05) according to Tukey’s Multiple Range Test.

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
