# Peer review of "Pork Liver Pâté Enriched with Persimmon Coproducts: Effect of In Vitro Gastrointestinal Digestion on Its Fatty Acid and Polyphenol Profile Stability"

_nutrients, 2021, doi:10.3390/nu13041332_

Round 1
Reviewer 1 Report
Interesting paper
Agrofood coproducts are used to enrich meat products to reduce harmful compounds and contributing to fiber and (poly)phenols. s/b Agrofood coproducts are used to enrich meat products to reduce harmful compounds and contribute to fibre and (poly)phenolic enrichment.
Could you expand your rationale - Why was pork pate selected as the food product of choice, given its basic composition it cannot be considered to be a healthy food nor have claims associated with it?
You mention nitroso and lipid oxidation compounds yet there is no measures within the paper, except for TBARs. You have no bioactivity measures of nitrous compounds of fermented i.e cytotoxicity to colon cells, inflammatory changes, cellular stress, DNA damage or the like. Therefore you may need to alter your introduction to align with the the lipid focus. Without changes in bioactivity you can only comment on composition, had you shown any cellular changes you could draw more readily up on a disease health discussion.
Articulate limitations of using UV based detection + retention time rather than an MSn.
What is rationale for persimmon specifically other fruits also have these capacities, or are they just a model fruit rich in PPs?
Tuckey post host test S/B Tukey's post host test.
Author Response
We thank all your comments and suggestions that allow us to clarify the message of our paper.
The paper has been carefully revised and language and grammatical errors have been corrected.
I am going to answer all your comments point by point. Your comments are in blue and our answers in black color.
Agrofood coproducts are used to enrich meat products to reduce harmful compounds and contributing to fiber and (poly)phenols. s/b Agrofood coproducts are used to enrich meat products to reduce harmful compounds and contribute to fibre and (poly)phenolic enrichment.
Ok. The sentence has been changed.
Could you expand your rationale - Why was pork pate selected as the food product of choice, given its basic composition it cannot be considered to be a healthy food nor have claims associated with it?
Pork liver pâté is a well-known meat product with a high amount of proteins and iron with high bioavailability. In addition, it is also an excellent source of other micronutrients such as zinc, and vitamins. It is a cheap product and so, widely consumed and especially recommended for children and elderly. It is true that it has a high content in fats and for this reason, we consider that is an interesting strategy to work with it in view of increasing its healthiness, and especially for this study in which we aimed to assess the lipid oxidation (TBARs) and the stability of the fatty acid profile, during in vitro gastrointestinal digestion. For all these reasons pâté has been selected.
You mention nitroso and lipid oxidation compounds yet there are no measures within the paper, except for TBARs. You have no bioactivity measures of nitrous compounds of fermented i.e cytotoxicity to colon cells, inflammatory changes, cellular stress, DNA damage or the like. Therefore you may need to alter your introduction to align with the lipid focus. Without changes in bioactivity you can only comment on composition, had you shown any cellular changes you could draw more readily up on a disease health discussion.
It is true that we don’t show here nitrite residual level in pâté sample but they have been previously published and referenced in this article. It is not the same case for cytotoxicity analysis because we have not experience in that.
However, we consider that the short paragraph that we have included in the introduction section (page 1, lines 37-41) about nitroso compounds is very appropriate because nitrite is used in the formula as a preservative and so the risk of nitrosamines development.
Regarding the discussion section, we hypothesized that polyphenols that remain join to bound cell could be metabolized by colonic bacteria, and we reported some studies in relation to the effects of polyphenols on colon diseases, but we don´t say that our study support this idea because, as you have commented, more determinations are needed for confirming this association.
Articulate limitations of using UV based detection + retention time rather than an MSn.
You have right that the better would have been to use MSn, but we have not this facility. In any case, as regards HPLC, we also used the absorbance spectrum, and extensive work has been made to identify these compounds. A lot of works have been previously published using these tools for polyphenols identification.
What is rationale for persimmon specifically other fruits also have these capacities, or are they just a model fruit rich in PPs?
We have a lot of experience working with agro-industrial co-products, most of them rich in polyphenols and we suppose that persimmon could be considered as a model fruit rich in PPs. However, this is the first study in which such a deep polyphenol characterization and analysis after in vitro digestion have been made. In this case, the effect of food matrix is also very important
Tuckey post host test S/B Tukey's post host test.
Ok. The sentence has been changed.

Reviewer 2 Report
The study of Lucas-González et al. examined the stability of free and bound (poly)phenols in pork liver pâtés enriched with persimmon flour before and after in vitro gastrointestinal digestion and the effect of lipolysis in the mentioned determinations using two pancreatins with different lipase activity. The study is well-designed and all the experiments are thorough. However, many changes concerning text editing and English language are needed. Listed below there are some examples.
Introduction
- Page 1, line 41: add “such”
Materials and Methods
- Page 3, line 119: remove “to”
- Page 4, line 149: “and” instead of “an”
- Page 4, line 171: “in the dark” instead of “under dark”
- Page 4, line 181: is the word “passed” missing?
- Page 5, line 201: “lyophilized” instead of “lyophilizate”
- Page 5, line 222: “Methylation” instead of “Mehylation”
- Page 5, line 228: “were” instead of “was”
- Page 6, line 266: “were” instead of “was”
Results and discussion
- Page 7, line 282-283: “Figure 2 shows the absorbance spectrum of some of these compounds” instead of “In Figure 2, it can be observed the absorbance spectrum of some of these compounds.”
- Page 7, line 291: remove “to”
- Page 7, line 293: “as it can” instead of “as can”
Author Response
We thank all your comments and suggestions that allow us to clarify the message of our paper.
The paper has been carefully revised and language and grammatical errors have been corrected.
All your grammatical errors detected have been changed.
